# Hydrogen Diffusion on, into and in Magnesium Probed by DFT: A Review

Marina G. Shelyapina

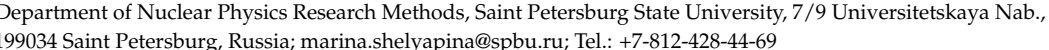

Department of Nuclear Physics Research Methods, Saint Petersburg State University, 7/9 Universitetskaya Nab., 199034 Saint Petersburg, Russia; marina.shelyapina@spbu.ru; Tel.: +7-812-428-44-69

**Abstract:** Hydrogen is an energy carrier that can be a sustainable solution for alternative energy with zero greenhouse gas emissions. Hydrogen storage is a key point for hydrogen energy. Metals provide an access for safe, controlled and reversible hydrogen storage and release. Magnesium, due to its outstanding hydrogen storage capacity, high natural abundance, low cost and non-toxicity is one of the most attractive materials for hydrogen storage. The economic efficiency of Mg as a hydrogen accumulator is limited by its sluggish hydrogen sorption kinetics and high stability of its hydride $MgH_2$. Many attempts have been made to overcome these shortcomings. On a microscopic level, hydrogen absorption by metal is a complex multistep process that is impossible to survey experimentally. Theoretical studies help to elucidate this process and focus experimental efforts on the design of new effective Mg-based materials for hydrogen storage. This review reports on the results obtained within a density functional theory approach to studying hydrogen interactions with magnesium surfaces, diffusion on Mg surfaces, into and in bulk Mg, as well as hydrogen induced phase transformations in $MgH_x$ and hydrogen desorption from $MgH_2$ surfaces.

**Keywords:** metal–hydrogen systems; magnesium; $MgH_2$; hydrogen storage; absorption; desorption; diffusion; surface interactions; activation energy; diffusion paths; vacancies; density functional theory





## 1. Introduction

Hydrogen is considered an energy carrier [1–3], which can be a sustainable energy supply solution with zero emission of greenhouse gasses. Metals, besides having an advantage of very high hydrogen density as hydrides, provide an option for safe, controlled and reversible hydrogen storage and release [1,4,5].

To be a "good" hydrogen storage material, a metal–hydrogen system must meet several criteria that include high hydrogen storage capacity (measured in weight percentages, wt.%), fast hydrogen sorption kinetics, low hydrogen release temperature, reversibility of the hydrogen absorption–desorption reaction, good cycling life, low cost, etc. To date there is no material that would satisfy all these requirements.

Magnesium, due to its outstanding hydrogen storage capacity up to 7.6 wt.% for $MgH_2$, high natural abundance, low cost and non-toxicity, is considered one of the most attractive materials for the hydrogen storage application and is already used in practice [6,7]. However, its rather slow hydrogen absorption and desorption kinetics [8] as well as high dissociation temperature (above 673 K) [9,10] and the noticeable reactivity toward air (oxygen) impede the economic efficiency of Mg-based hydrogen storage devices.

Recently, many attempts have been made to overcome these drawbacks [11,12]. Most of them comprise the destabilization of Mg or Mg-based metal hydride systems by the introduction of defects [11] in several ways, such as mixing Mg or $MgH_2$ with transition metals (TM) [13–15] or their oxides [16,17], nanostructuring of Mg using high energy ball milling [18–20] or severe plastic deformation [18,20,21] or ion irradiation [22,23].

On a microscopic level, hydrogen absorption by metal is a complex multistep process that includes the dissociation of the $H_2$ molecule on the metallic surface and stepwise

diffusion of atomic hydrogen into and further in bulk metal, hydride phase nucleation and growth, etc. Hydrogen desorption likewise comprises surface desorption, hydrogen atom or vacancy diffusion, phase nucleation and growth, etc. To follow this whole process experimentally is an impossible task. Experiments provide access either to a dynamic process upon (de)hydrogenation, or to the initial and final states of the substance in an equilibrium state. However, even in situ experiments are limited by a dramatic mismatch between hydrogen jump rates responsible for diffusion and the experimentally accessible timescale. Compared to many other atoms in solids, hydrogen in metals exhibits rather fast diffusion [24]. This entails applications of a large variety of experimental methods, including electrochemistry, mechanical relaxation, nuclear magnetic resonance (NMR), quasi-elastic neutron scattering (QENS), perturbed angular correlation, and others [24–26].

It should be noted that different experimental techniques explore different diffusion processes. For example, permeation techniques involve studies of hydrogen transport diffusion into a bulk: hydrogen concentration changes during an experiment, and H diffusion can be described by the Fick's law. NMR probes hydrogen jumps from one interstitial site to another, which results in a translational motion, i.e., diffusion, but the hydrogen concentration globally does not change. The diffusion of hydrogen in this case is described using the self-diffusivity, which is related to the motion of a tagged molecule (atom) by the well-known Einstein relation [27].

Several techniques, such as NMR or QUENS, can probe atomic jump rates in a wide range. For example, NMR provides access to jump rates from $10^{-1}$ to $10^{11}$ s$^{-1}$ [28], which makes it a powerful tool for studying hydrogen self-diffusion in metals [28–31]. In addition, in an experiment we survey the statistical ensemble behavior. Being very effective at measuring both the hydrogen diffusivity and jump frequency, experimental methods cannot describe microscopic hydrogen migration processes and steps in the hydride formation, although this knowledge is essential for the development of a strategy for the development of new materials with specified parameters. From this perspective, theoretical calculations are very helpful, especially in combination with experimental studies.

A comprehensive theoretical study of hydrogen sorption by metals involves the estimation of the adsorption and absorption energies, subsurface barriers of hydrogen penetration in metal, activation barriers for hydrogen diffusion between interstitial sites, hydrogen jump rates, etc. Density functional theory (DFT) is a powerful tool that provides access to these parameters. To date, a huge amount of data on the study of various metal–hydrogen systems by DFT, including magnesium–hydrogen systems, has already been accumulated. In this review, an attempt is made to describe the recent results of the DFT studies of hydrogen interaction with magnesium surfaces, hydrogen diffusion on Mg surfaces, into and in bulk Mg, as well as hydrogen induced phase transformations in MgH$_x$ and hydrogen desorption from MgH$_2$ surfaces. However, first it is necessary to give a short introduction, with a brief description of hydrogen's interaction with metal and to introduce the basic concepts and approaches used to calculate hydrogen diffusion paths, activation barriers and jump rates within the framework of DFT.

## 2. Interaction of Hydrogen with Metals

Hydrogen is very reactive and combines with almost every element in the periodic table except for noble gases exhibiting a variety of chemical bounding [5]: ionic, covalent, metallic or their combination. Alkali and alkaline earth metals, such as Li, Ca and Ba form ionic hydrides. Their electronegativity is noticeably smaller than the hydrogen one, and hence hydrogen acts as an oxidizing agent. These hydrides generally are very stable. Transition metals normally form metallic hydrides and their stability may vary in a wide range [5]. Electronegative elements, such as B and Al, form covalent hydrides often with complex anion, [BH$_4$]$^-$, [AlH$_4$]$^-$, or polymeric structures. Magnesium hydride represents an intermediate case: a mixture of covalent and ionic bonding [32,33].

The process of the hydride formation is characterized by a gradual dissolution of hydrogen in a host metal *M* (solid solution *M*-H), followed by phase transformations with

the appearance of a hydride of certain stoichiometry. Very often the reaction is reversible but causes embrittlement of metal. The reaction of hydride formation can be represented as follows:

$$M + \frac{x}{2}\mathrm{H}_2 \leftrightarrow M\mathrm{H}_x + \Delta H, \tag{1}$$

where $M\mathrm{H}_x$ is a hydride of metal $M$ with hydrogen concentration $x = \mathrm{H}/M$, $\Delta H$ is the enthalpy of formation. If at normal conditions the reaction is exothermic, the corresponding hydride is stable, if endothermic, the hydride is unstable. For $\mathrm{MgH}_2$ $\Delta H = -75.7$ kJ/mol $\mathrm{H}_2$ [34] that determines the high temperature of its hydrogen release.

Interaction of hydrogen from gas phase with a metal is illustrated in Figure 1. It is a complex process that includes following steps: (i) approaching $\mathrm{H}_2$ molecules to the metal surface under an external pressure; (ii) attraction of $\mathrm{H}_2$ molecules to the metal surface due to the van der Waals interaction (physisorption); (iii) dissociation of physisorbed $\mathrm{H}_2$ molecules on the metal surface under external impact (temperature and/or pressure); (iv) penetration of H atoms into the host metal lattice (chemisorption) and occupation of interstitial sites in the sub-surface layer; (v) hydrogen diffusion into the metal lattice with formation, first, a solid solution M-H, $\alpha$-phase (H/M < 0.1) and then a hydride $M\mathrm{H}x$, $\beta$-phase. The energy diagram for this multistep process is schematically shown in Figure 2.

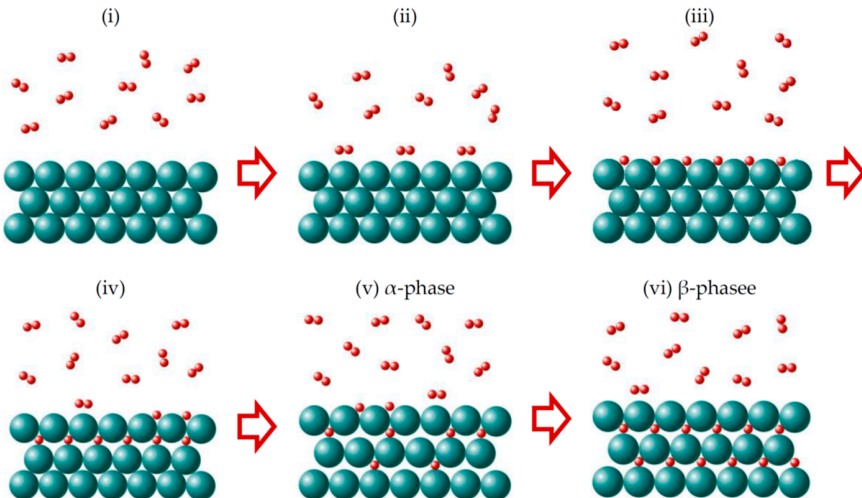

**Figure 1.** Interaction of hydrogen with metal. Adapted with permission from Shelyapina, M.G. *Metal hydrides for energy storage*; Springer Nature Switzerland AG, 2019 [5].

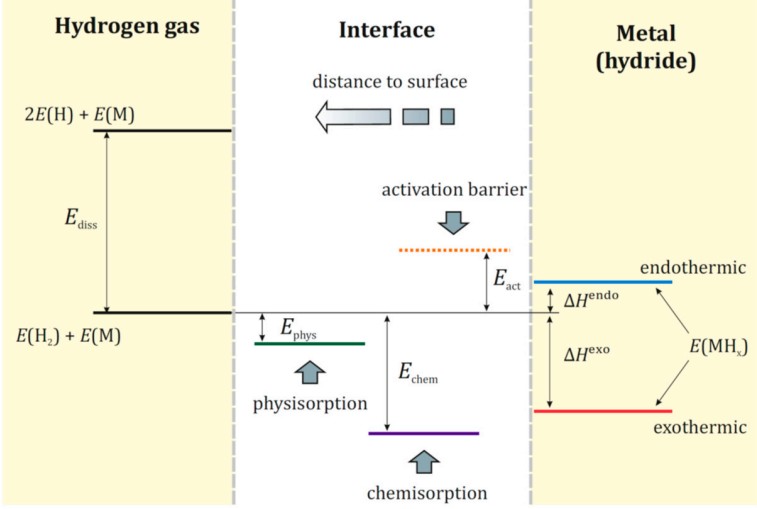

**Figure 2.** Energy diagram for hydrogen interaction with metal. Reproduced with permission from Shelyapina, M.G. *Metal hydrides for energy storage*; Springer Nature Switzerland AG, 2019 [5].

To be absorbed by metal, hydrogen molecules must dissociate. High-index Pd, Pt and Ni surfaces exhibit spontaneous $H_2$ splitting (non-activated hydrogen absorption) [29]. To make $H_2$ dissociate on an Mg surface, an extra portion of energy—the so-called activation energy $E_{act}$—has to be supplied. The origin of this activation barrier is mainly determined by electronic factors [30–33], but the hydrogen behavior is also influenced by an oxide film on the metal surface. In practice, to achieve better hydrogen capacity and kinetics before first hydrogenation, an activation is required. The activation normally involves internal cracking of metal particles to increase reaction surface area [34].

A detailed description of the hydrogen diffusion into metal can be found in reviews by Völkl and Alefeld [35] and Fukai [36]. In a metallic lattice, hydrogen atoms may occupy tetrahedral (T) and octahedral (O) interstitial sites (see Figure 3). The preference in the interstitial site occupancy depends not only on the host metal, but on the hydrogen concentration as well. Hydrogen, penetrating into metal, causes the volume expansion of the host lattice that can be followed by several structural phase transformations.

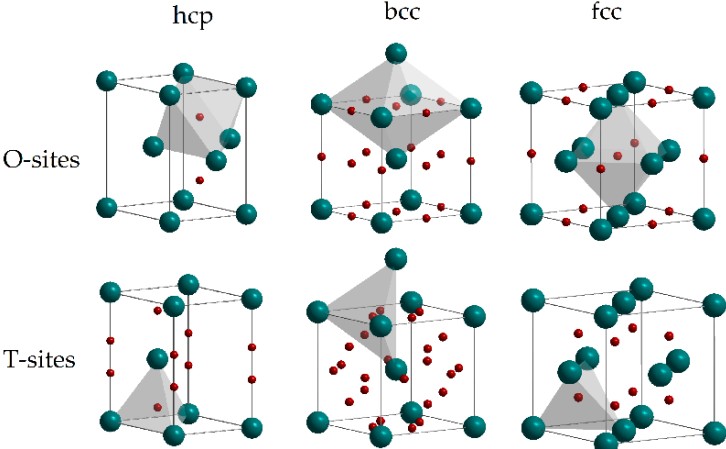

**Figure 3.** Octahedral (O) and tetrahedral (T) interstitial sites in hcp, bcc and fcc lattices. Reproduced with permission from Shelyapina, M.G. *Metal hydrides for energy storage*; Springer Nature Switzerland AG, 2019 [5].

## 3. DFT Method to Study Hydrogen Diffusion

### 3.1. Potential Energy Surface and Activation Energy

A description of the DFT methods can be found anywhere [35]. The result of modeling (the quality of calculations) depends on the parameters of the method: the choice of the exchange–correlation functional, the choice of basic functions (or cutoff energy for the basis set), the k-point mesh, accounting or neglecting atomic vibrations, the choice of pseudopotential (if appropriate), etc. The above affects not only the absolute values of the total energy of the system, but also the equilibrium geometry and three-dimensional potential energy surface (PES). This should be taken into account when comparing the results of calculations obtained within various realizations of DFT.

Modeling the diffusion of hydrogen on a metallic surface or inside a bulk metal within the framework of DFT is associated with the problem of PES construction and determination of minimal energy path (MEP). To diffuse between two sites (interstitial/interstitial or surface/subsurface/interstitial), which are local minima on PES, a hydrogen atom needs to pass a transition state (TS) and overcome an energy barrier $\Delta E$. This transition state is not unique, and the search for the optimal TS is in fact a search for the lowest saddle point at the edge of the PES basin of the initial state.

Hydrogen diffusion paths can be calculated with the climbing image nudged elastic band (CI-NEB) method [36–38], which allows us to find MEP in an accurate and fast way. The force acting on the hydrogen at any point along the path is directed along the path, and the energy is stationary for any perpendicular displacement (see Figure 4). Activation energies are maxima on the MEP, which are saddle points on PES. Path optimization

is performed using an iterative procedure; at each iteration all images are moved and evaluated. Usually, calculations are limited to 10–15 images, since the procedure is rather costly even to describe the diffusion of a single particle. Acceleration of the rate of NEB convergence can be achieved by Gaussian process regression (GPR) machine learning methods [39–41].

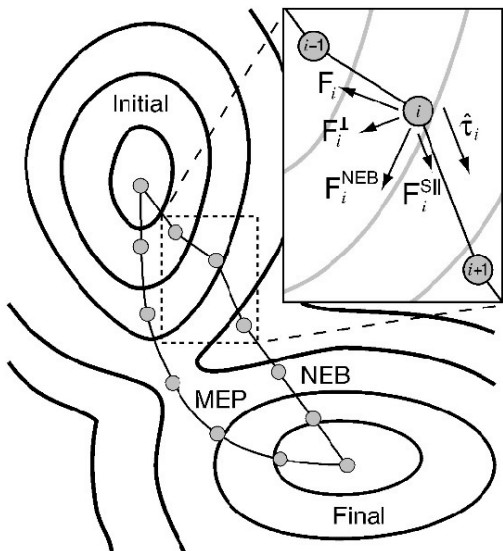

**Figure 4.** Schematic diagram to show the forces and their relation to MEP: the total force on each image $\mathbf{F}_i^{\mathrm{NEB}}$ is the sum of the true force perpendicular to the local tangent $\mathbf{F}_i^{\perp}$ and the spring force along the local tangent $\mathbf{F}_i^{\mathrm{S}\|}$. Reproduced with permission from Sheppard, D. et al. *J. Chem. Phys.*; published by American Institute of Physics, 2008 [42].

*3.2. Hydrogen Jump Rate and Diffusion Coefficient*

Within the framework of a DFT approach, the hydrogen diffusion coefficient in the metal lattice can be estimated using the well-known expression proposed by Werth and Zener [43]:

$$D = nL^2\Gamma, \tag{2}$$

where $n$ is a numerical coefficient that depends on the symmetry and location of the interstitial site occupied by hydrogen, $L$ is the projection of the jump length on the direction of diffusion. $\Gamma$ determines the hydrogen jump rate between adjacent interstitial sites and can be expressed in terms of vibrational frequencies at the transition and ground states [44].

For magnesium hydride below hydrogen release temperature the following ratio for the phonon energy is fulfilled, $h\nu > kT$. Phonon frequencies that belong to the vibrations of hydrogen atoms in $MgH_2$ are normally above 600 cm$^{-1}$ [45]. Within this low temperature approximation the hydrogen jump rate can be written as follows [44,46]:

$$\Gamma = \frac{k_{\mathrm{B}}T}{h} \exp\left[-\frac{\Delta E + \Delta \mathrm{ZPE}}{k_{\mathrm{B}}T}\right]. \tag{3}$$

Here, $\Delta E$ is the activation barrier that the hydrogen atom needs to overcome to pass from one interstitial site to another, and $\Delta \mathrm{ZPE}$ is the difference in zero-point energy (ZPE) contributions between the ground and transition states. By substituting Equation (3) into Equation (2), one obtains the following expression for the diffusion coefficient:

$$D = n\beta L^2 \frac{k_{\mathrm{B}}T}{h} \exp\left[-\frac{\Delta E + \Delta \mathrm{ZPE}}{k_{\mathrm{B}}T}\right]. \tag{4}$$

The numerical coefficient $n$ for any crystal lattice can be easily evaluated according to Zener [47]. The parameter $\beta$ is introduced to account for the dependence of the diffusion

coefficient on the hydrogen concentration. However, Equation (4) is valid only for a single-stage hydrogen diffusion path. If there is a local energy minimum on the diffusion path, hydrogen can linger in a metastable state before making a further jump forward or back (see Figure 5). To take into account complex diffusion paths, it is necessary to consider the forward and reverse hydrogen flows between stable interstitial sites. For more details see Ref. [44].

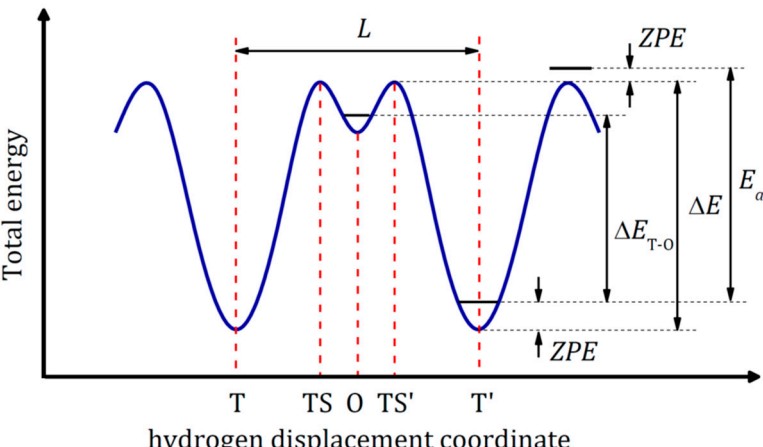

**Figure 5.** Energy profile for an indirect hydrogen diffusion path with a local metastable state on the hydrogen diffusion path. Reproduced with permission from Klyukin, K. et al, *J. Alloys Compd.*; Elsevier B.V., 2015 [48].

Assuming the constant gradient of the particle flux density and applying Fick's law, on can obtain an expression for a hydrogen diffusion between two stable sites through a metastable one. For example, for the hydrogen diffusion in bcc lattice between two T-sites through a metastable O-site the diffusion coefficient can be written as follows:

$$D'_{bcc} = 2\beta L^2 \Gamma_T \left( 1 + \frac{1}{2} e^{-\frac{\Delta E_{T-O}}{k_B T}} \right)^{-1}. \tag{5}$$

### 3.3. Hydrogen Solubility and Hydrogen Vacancy Energies

In metal–hydrogen systems, $MH_x$, the hydrogen diffusion coefficient decreases with hydrogen concentration $x$: H atoms penetrating into metal lattice occupy more and more interstitial sites making unavailable a part of diffusion paths. The simplest way to account for $x$ is to weight the diffusion coefficient by the probability $\beta$ that the target position is effectively accessible [49]. This parameter can be easily evaluated for regular lattice of pure metals, but for substituted lattices distribution of hydrogen over possible interstitial sites must be calculated in an accurate way. The distribution of H atoms over interstitials is given by a Fermi–Dirac distribution [50,51], derived in terms of general thermodynamic description:

$$c^{(i)} = \frac{c \cdot P^{(i)}}{1 + \exp\left[ \left( E_{sol}^{(i)} + f(c) - \mu \right) / RT \right]}, \tag{6}$$

where $c^{(i)}$ is the hydrogen occupation on a site type $i$; $c = \sum_i c^{(i)}$ is the total hydrogen site occupancy; $E_{sol}^{(i)}$ is the hydrogen solubility energy (the energy that costs to remove a hydrogen atom from a certain interstitial site; $f(c)$ is the long-range effective hydrogen-hydrogen interaction (normally, it is assumed that this H–H interaction depends on the total H concentration $c$ only [51]); $P^{(i)}$ is the probability of finding a site type $i$ for a given alloy composition; $\mu$ is the chemical potential of a hydrogen atom in metal; $R$ and $T$ are the gas constant and the absolute temperature, respectively.

In hydrides, the hydrogen chemical potential, $\mu$, is normally assumed to be identical to that of the gas phase [24,52]. As soon as in DFT calculations H–H repulsion is taken into account at the level of the Hamiltonian description, only the hydrogen solubility energy and chemical potential need to be evaluated. To consider all possible types of hydrogen environment and/or to increase calculation accuracy for minimizing effect from adjacent unit cells a supercell approach is usually applied. For an unsubstituted lattice of a hydride $MH_2$ the hydrogen solubility energy $E_{sol}$ can be calculated as:

$$E_{sol} = E_{MH_2} - \left[ E_{MH_2-xH} + \frac{x}{2} E(H_2) \right]. \tag{7}$$

Here, $x$ is a hydrogen vacancy concentration created by eliminating one hydrogen atom. For substituted or disturbed lattices, one has a set of $E_{sol}^{(i)}$ corresponding to inequivalent hydrogen interstitials. To obtain relevant solubility energy, the total energy of the hydride without and with hydrogen vacancy ($E_{MH_2}$ and $E_{MH_2-xH}$, respectively) as well as the total energy of the $H_2$ molecule $E(H_2)$ must be calculated using the same theory level, e.g., accounting for ZPE or not (see Section 3.4). Such an approach was successfully applied to study hydrogen distribution and interstitial diffusion in disordered Ti-V-Cr alloys [52,53]. Note that the hydrogen solubility energy in Equation (7) taken with the opposite sign is the energy for a single H-vacancy formation.

### 3.4. Zero-Point Energy

ZPE takes into account atomic vibrations that, due to the uncertainty principle, exist even at $T = 0$. It can be estimated as $\sum_i \frac{h\nu_i}{2}$, where the sum runs over all vibrational frequencies of normal modes. Its contribution may be very important and correct calculations of the activation barrier assume accounting for ZPE (see Equations (3) and (4)). Its calculation requires construction of the phonon spectrum. Within DFT phonon frequencies can be obtained either by direct calculations or using the linear-response approach. In the direct method the phonon energy is computed as a function of the displacement amplitude in terms of energy difference of the distorted and the unperturbed lattices, the so called frozen-phonon method [54]. It works well if the phonons wavelength is compatible with the periodic boundary conditions applied to the considered cell. More accurate approaches require calculations of the forces-constant matrices [54–58] that is very time-consuming.

The linear response method allows us to reduce the computational effort [59,60]. In this method the dynamical matrix is expressed in terms of the inverse dielectric matrix, which describes the response of the valence electron density to a periodic perturbation of the lattice. It is fast and accurate for studying metals [61–65] and even for metal–hydrogen systems provides satisfactory results [48,52,66–68].

## 4. DFT Modelling of Mg-H Systems

### 4.1. Mg-H Bonding and Effect of TM Substitution

Both the hydride stability and hydrogen sorption processes are primarily governed by metal–hydrogen bonding. According to theoretical studies, bonding in $MgH_2$ is a complex mixture of ionic and covalent contributions that determines the rather high thermodynamic stability of $MgH_2$ [33,69,70]. Partial substitution of TM for Mg weakens the bonding between magnesium and hydrogen [11,70–74], but the TM-H bonding appears rather strong: TM $d$-states are strongly hybridized with $s$-states of the hydrogen atoms [70,74]. Substitution effects are normally accompanied by local distortions that can be modeled within a cluster approach. In general, the main conclusions obtained to study effects of TM doping on $MgH_2$ within the cluster approach [71,73,75] are in line with supercell periodic boundary calculations [11,70–74].

As was shown experimentally the stability of the complex $Mg_7MH_x$ and $Mg_6MH_x$ hydrides with $M$ = Ti, V, Nb is lower compared to $MgH_2$ [76–78]. However, as soon as Mg does not form any alloys with these TMs, the structures are stabilized by hydrogen; no

binary compounds exist after hydrogen release. As it was shown by Shelyapina et al. [79] co-substitution Mg by Ti and Zn or Al leads to stabilization of the alloys with simultaneous decreasing stability of the corresponding hydrides due to the TM–H bond weakening.

### 4.2. Mg/TM Thin Films and bcc Mg Phase Stability

Besides, binary Mg-TM alloys may exist as thin films, e.g., Mg-Ti [80] or Mg-Nb [81]. DFT calculations suggest that in thin films Mg may adopt the bcc structure of Nb layers [82–84] resulting to an hcp-to-bcc phase transformation of Mg [85]. This phase transformation can be described by Burgers' model [86]. Within this model the hcp–bcc phase transition can be considered as the result of two process supposed to be independent from each other: a shear deformation from bcc (110) plane to the hexagonal basal plane and a slide along the [110] direction of planes (see Figure 6a). According to PES calculations, Figure 6b, during the hcp-to-bcc phase transformation in Mg at first the shear deformation dominates but at the end of transformation the slide displacement becomes stronger [85].

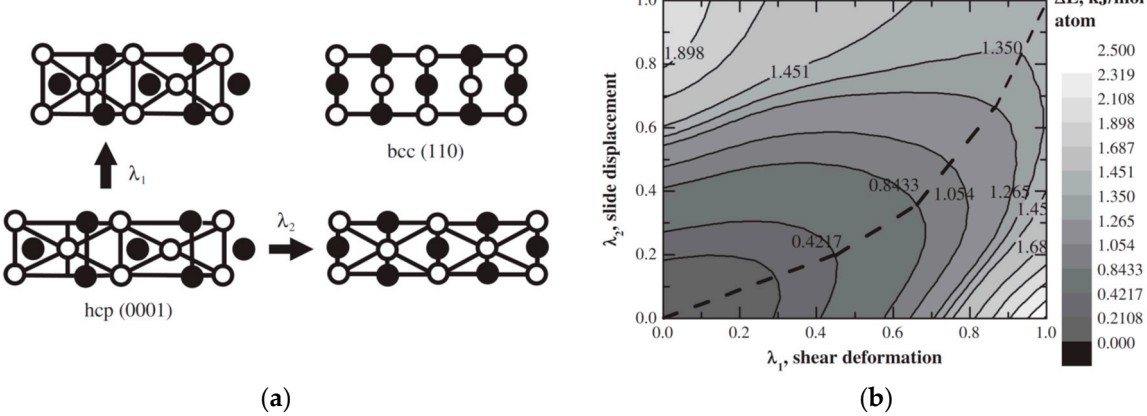

**(a)** **(b)**

**Figure 6.** Schematic representation (**a**) and PES counter plot (**b**) for the hcp-to-bcc phase transformation in magnesium (dashed line represents MEP). Reproduced with permission from Klyukin, K. et al, *J. Alloys Compd.*; Elsevier B.V., 2013 [85].

Despite bcc structure of Mg is metastable, and as it was shown by Klyukin et al. [48] even a small tetragonal distortion provokes the bcc-to-fcc transition, the bcc structure is a key for fast hydrogen kinetics in Mg. In more details this is discussed further in Sections 4.4 and 4.5.

### 4.3. Hydrogen Molecule Dissociation on Mg(0001) Surfaces and Migration into the Subsurface

Many theoretical efforts have been made to understand sluggish kinetics of hydrogen sorption by Mg. It is believed that the kinetics is limited either by poor dissociative chemisorption of $H_2$, or by a highly stable surface hydride film blocking the diffusion of atomic hydrogen into magnesium.

The dissociation of molecular hydrogen on a Mg(0001) surface and the subsequent diffusion of atomic hydrogen into the magnesium have been theoretically studied in several works [87–91]. The bridge site was found the most favorable for $H_2$ dissociation with an activation energy of about 1.4 eV and barrier for H atomic diffusion from surface to bulk of 0.15–0.53 eV (depending on the method of calculation and initial and final hydrogen sites). These results suggest that the dissociation and recombination of $H_2$ are the rate-limiting processes in ab- and desorption of hydrogen at the Mg(0001) surface [87].

Interactions between hydrogen and Mg surface is also complicated by presence of defects. As it was shown in Ref. [91] vacancy defects on the Mg(0001) surface decrease the dissociation barrier from 1.42 eV for the defect-free surface down to 1.28 eV, and also change the preferential diffusion entrance for hydrogen atom to the subsurface reducing the energy barrier for hydrogen diffusion. According to Han et al. [91] regardless of whether

on defect-free or vacancy defective Mg(0001) surfaces, the fcc channel is not the MEP of hydrogen atom diffusion from the surface into the magnesium bulk, as it was supposed in earlier studies [92], but a spiral channel composed by octahedral and tetrahedral interstitial sites, as shown in Figure 7.

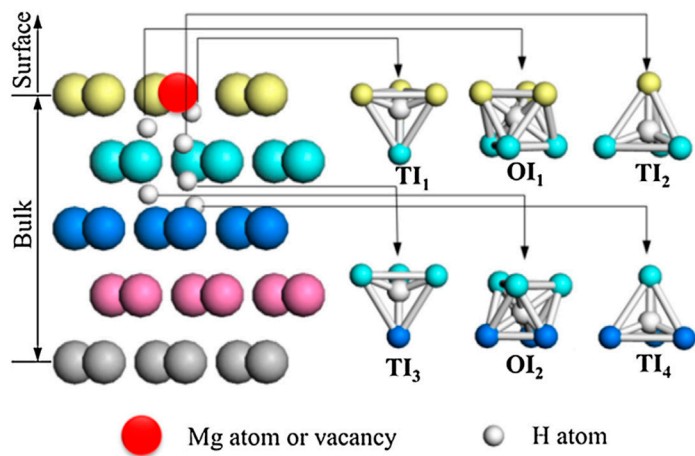

**Figure 7.** Schematic diagram of hydrogen atom diffusion in bulk Mg from the Mg(0001) surface. Different H interstitial sites with their local environment are shown. Reproduced with permission from Han, Z. et al. *Appl. Surf. Sci.*; Elsevier B.V., 2017 [91].

Attempts to understand mechanism of enhancement of hydrogen sorption kinetics in presence on TM additives have been undertaken in a number of theoretical studies [90,93–96]. In Ref. [93] it was shown that Ti and Pd dopant affects the reaction path oppositely. Pd dopant dramatically decreases activation barriers for both $H_2$ dissociation and atomic H diffusion on the Mg surface. On the Ti-incorporated surface, the $H_2$ molecule dissociates almost spontaneously, the activation barrier is rather low, but diffusion away from the Ti site is very unfavorable due to strong Ti-H bonding.

A systematical study of hydrogen dissociation and diffusion on the Mg(0001) surface doped by different TMs in the first layer (TM = Ti, Zr, V, Fe, Ru, Co, Rh, Ni, Pd, Cu, Ag) has been made by Pozzo et al. [96]. It was found that TMs on the left part of the periodic table eliminate the dissociation barrier, however, the strong interaction of hydrogen with a dopant prevents diffusion away from the catalytic center that would slow down hydrogen absorption. In contrast, TMs on the right side of the periodic table bind H atoms loosely, allowing them easily to diffuse away, but their effect on the activation barrier for the $H_2$ molecule dissociation is small: the barrier still exists and is high. However, such as Ni atom prefers to migrate inside Mg matrix rather than to occupy in or over the topmost Mg(0001) surface [95]. Being placed inside Mg bulk Ni not only does not improve $H_2$ dissociation but has a detrimental effect on atomic hydrogen diffusion [94,95], whereas Ni atom locating in/over the topmost Mg(0001) surface is an excellent catalyst for $H_2$ dissociative chemosorption. Theoretical studies of hydrogen dissociation and diffusion mechanism on the Mg(0001) surface co-doped with V and Ni have shown that the presence of V atom at the subsurface layer not only stabilizes Ni at the surface layer but also facilitates $H_2$ dissociation and hydrogen diffusion on and inside Mg [94].

It should be noted that all these studies do not account for ZPE. According to Jiang et al. [92], even for the pure Mg(0001) surface the absolute hydrogen absorption energies calculated by neglecting ZPE is overestimated by 0.05 eV/H. Often ZPE in Mg-H systems is neglected, being considered a constant offset [97,98]. A systematic study of hydrogen dissociative adsorption on 28 surfaces of 12 different TMs caried out in Ref. [99] proved that preferred hydrogen binding sites do not change upon correcting for the ZPE. Nevertheless, the ZPE contribution to adsorption energy is not negligible and is between 0.13 and 0.19 eV/H. This allows us to assume that in Mg with TM dopant, the contribution of ZPE to the hydrogen adsorption energy may be more meaningful than in pure Mg.

Besides TMs some non-metallic elements, such as carbon, promote activation of Mg and improve the hydrogen absorption kinetics [100,101]. As was shown theoretically by Du et al. [88] incorporated carbon atoms facilitate the dissociation of $H_2$ and the diffusion of atomic H on the Mg(0001) surfaces. A carbon atom on the Mg(0001) surface can easily migrate into the subsurface layer and occupy an fcc interstitial site. This process is accompanied by charge transfer from nearby Mg atoms to carbon, that (i) facilitates the dissociative chemisorption of $H_2$ on the Mg(0001) surface [88], and (ii) enhances the surface migration and subsequent diffusion of H atoms into the sub-surface layers, reducing the activation barrier by 0.16 eV [102]. MEP for this process for pure and C-doped Mg(0001) surface is shown in Figure 8. However, this result was obtained without accounting for zero-point vibrations. ZPE correction makes the difference between the activation barriers for pure and C-doped Mg(0001) surface less pronounced: 0.011 eV only, but as soon as the reaction paths are very similar in each case (see Figure 8), the inclusion of the ZPE correction does not affect the main conclusion concerning the effect of the carbon dopant on hydrogen adsorption on the Mg(0001) surface.

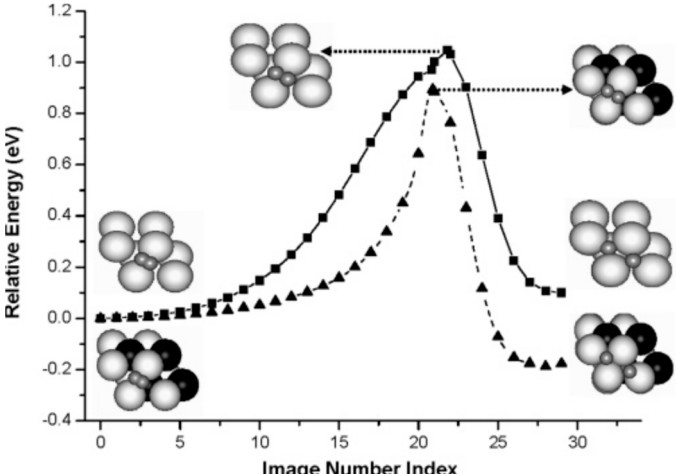

**Figure 8.** The energy profiles for $H_2$ molecule dissociation on a Mg(0001) surface: with (squares) and without (triangles) carbon substitution. The large gray, large black, and small gray balls represent Mg, C, and H atoms, respectively. Three special configurations, initial state, transition state, and final states along the MEPs are shown. Reproduced with permission from Du, A.J. et al. *J. Phys. Chem. B*; American Chemical Society, 2006 [102].

### 4.4. Hydrogen Induced Phase Transformations in MgH$_x$

In previous section we discussed hydrogen diffusion into hcp Mg. However, the hcp structure of MgH$_x$ exists in a very narrow hydrogen concentration range [103]. Hydrogen charging results in volume expansion of the metallic lattice with subsequent phase transformations and formation eventually $MgH_2$. It is impossible to survey experimentally these transforms upon hydrogenation; only the initial and final states can be caught. From this point of view, theoretical calculations are a unique tool for studying this process.

When penetrating into the hcp Mg lattice hydrogen may occupy O or T interstitial sites (see Figure 3). Interstitials, that hydrogen prefers to occupy in Mg, remain the subject of discussion [48,85,87,104–106]. According to Klyukin et al. [48] at very low hydrogen concentrations, T-sites in hcp Mg are more favorable with energy gain of 0.08 eV, but with the hydrogen concentration increasing O-sites become more stable.

Hydrogen induced stepwise phase transformations from hcp Mg to rutile $MgH_2$ were theoretically studied by Klyukin et al. [85]. Heat of formation for various phases of MgH$_x$ calculated without ZPE contribution, are shown in Figure 9. According to calculations at $0.1 < x < 1.5$ the fcc structure is more energetically favorable. At $x \approx 1.5$ an fcc-to-rutile phase transformation occurs that is accompanied by atomic bonding change from metallic to ionic-covalent. It should be noted that a metastable fcc-$MgH_2$ phase was found

experimentally very recently [107]. Being obtained upon cold rolling of reacted ball milled MgH$_2$ it exhibits lower stability compared to rutile and better hydrogen desorption kinetics.

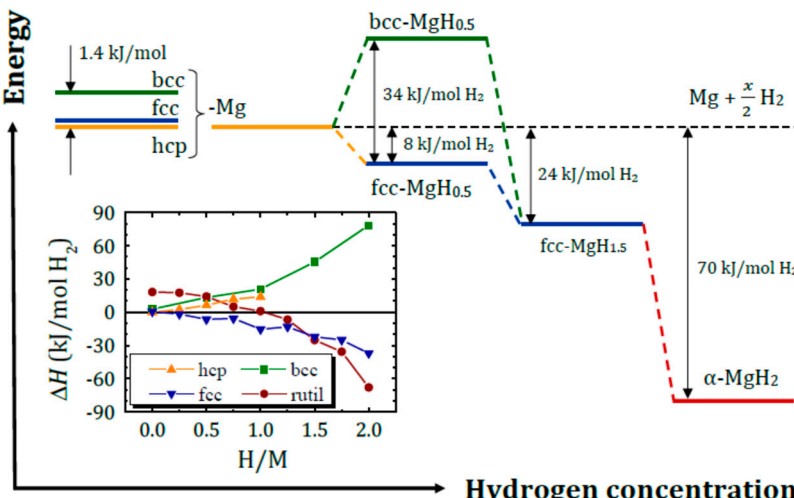

**Figure 9.** Energy diagram for hydrogen induced phase transformations in Mg; insert: heat of formation for various phases of MgH$_x$. Reproduced with permission from Grbović Novaković, J. et al. *ChemPhysChem*; Wiley-VCH Verlag GmbH & Co. KGaA, Weinheim, 2019 [11].

The bcc-MgH$_x$ structure is unstable within the whole hydrogen concentration range; however, in absence of hydrogen [82–84] or at low hydrogen concentration [85] it can be stabilized by presence of an adjacent layers of a bcc metal, such as Nb, V or Ti-V-Cr disordered alloys [52,53].

DFT calculations of the hydrogen site solubility in various MgH$_x$ structures reveal that for all the studied systems [85,106] except bcc MgH$_x$ [85] hydrogen atoms occupy adjacent interstitial sites forming a hydride layer that prevents hydrogen propagation into the Mg bulk. This so-called *blocking layer effect* is absent in bcc-Mg, in which hydrogen is spread over the lattice without forming clusters, that may explain acceleration of hydrogen sorption kinetics in Mg/Nb multilayers [81,108,109] and Mg@Nb (or Mg@V) composites [6,14,110,111].

### 4.5. Modelling of Hydrogen Diffusion in Bulk MgH$_x$

Despite multiple theoretical studies of hydrogen diffusion on the Mg surface and into the Mg bulk, calculations of hydrogen diffusion in bulk MgH$_x$ are not very numerous. Hydrogen diffusion coefficient in hcp-Mg has been theoretically calculated in Refs. [30,48,87,104] exhibiting a wide variation in two orders of magnitude. Klyukin et al. [30,48] studied hydrogen diffusion in hcp-, bcc- and fcc-MgH$_x$ with low ($x = 0.0625$) and moderate ($x = 0.5$) hydrogen concentrations. The activation energies for all possible diffusion paths were calculated using CI-NEB approach accounting for ZPE. The results are summarized in Figure 10a. It is worth noting that ZPE does not contribute much for MgH$_x$, less than 5% [48], but is of extreme importance for correct calculations of TM-H systems [30,52,53,112].

For hcp-MgH$_x$ the minimal activation energy was found for a T1→T2 hydrogen jump; however, accounting that diffusion assumes translational motion, in hcp Mg the following multistep hydrogen diffusion path is realized: O2→T2→T1→O1→T1′→ ... (where T1′ site is situated in an adjacent unit cell). The activation energy for this pathway is in fair agreement with experimental 0.25 eV [113]. In both bcc- and fcc-MgH$_x$ the hydrogen atoms prefer to occupy T-sites. The hydrogen diffusion paths were found T1→T3 (or T1→O1→T2, where hydrogen atoms move between two adjacent T-sites via an intermediate O-site) for the bcc lattice and T1→O1→T4 for the fcc one. Among considered structures bcc-MgH$_x$ exhibits the lowest activation energy for translational hydrogen motion.

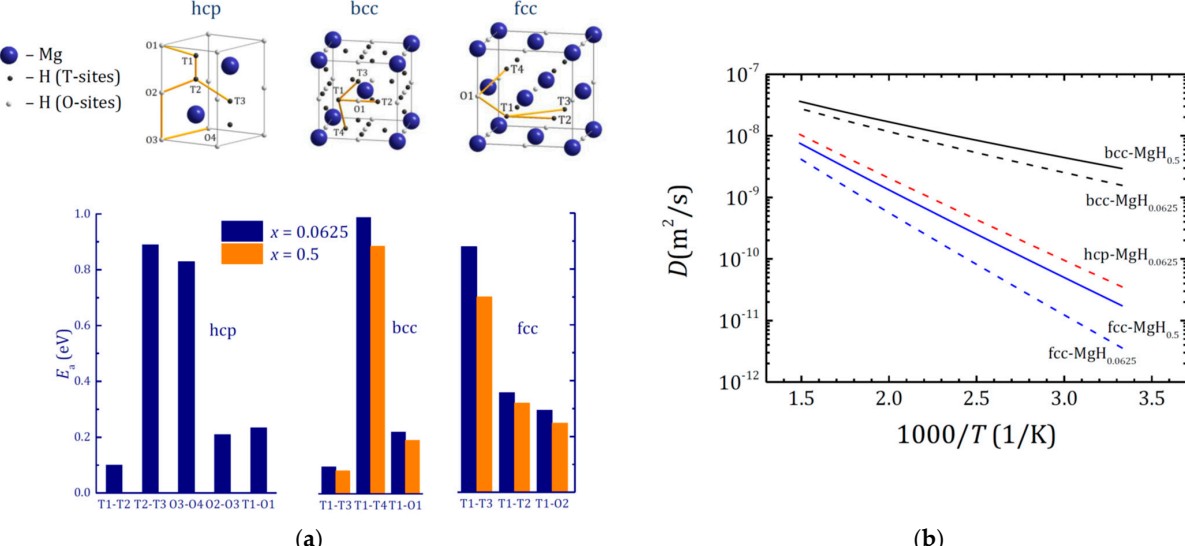

**Figure 10.** (**a**)—Possible H diffusion pathways in hcp, bcc and fcc MgH$_x$ and corresponding activation energy values; (**b**)—temperature dependence of the H diffusion coefficient $D$ calculated for the most favorable diffusion pathways in the hcp, bcc and fcc Mg lattices. Reproduced with permission from Klyukin, K. et al, *J. Alloys Compd.*; Elsevier B.V., 2015 [48].

The diffusion coefficient at 673 K for hcp MgH$_x$ with low hydrogen concentration calculated using equations like Equations (4) and (5) was found equal to $1.11 \times 10^{-8}$ m$^2$/s that agrees well with the experimental value of $2.07 \times 10^{-8}$ m$^2$/s obtained by Nishimura et al. using gas permeation technics [113] and other experimental studies of hydrogen diffusion in Mg (see Ref. [114] and references therein). Expanding this method to bcc- and fcc-MgH$_x$, it was found that bcc lattice exhibits the highest diffusion coefficient, whereas fcc shows the lowest one (see the calculated temperature dependence of hydrogen diffusion coefficient $D$ for various MgH$_x$ lattices in Figure 10b).

Since bcc MgH$_x$ does not share the blocking layer effect [85] it was supposed that this structure should promote fast hydrogen diffusion via the nucleation process [48]. The presence of TM layers or particles on the surface of Mg grains should help to locally stabilize the bcc-Mg arrangement, in which hydrogen diffusion occurs faster due to a lower activation energy and absence of the blocking layer effect.

### 4.6. Hydrogen Desorption from (001) MgH$_2$ Surfaces

The hydrogen desorption is primarily governed by Mg–H bonding, that was discussed in Section 4.1. Hydrogen desorption from low index MgH$_2$ rutile surfaces, MgH$_2$(001) and MgH$_2$(110) was theoretically studied by Du et al. [98]. They found that the MgH$_2$(110) surface is more stable than MgH$_2$(001), that can be perceived in term of bonds cut for Mg atom on the surface: two bonds for MgH$_2$(001) and only one for MgH$_2$(110). This conclusion is supported by experimental observations for thin films [115,116] and more resent calculation of a wider set of MgH$_2$ surfaces [117].

The activation barrier for hydrogen desorption depends not only on the surface index, but on the specific surface sites for the recombinative hydrogen atoms. The lowest activation energy, 1.78 eV, was found for desorption of one bridging and one in-plane H atom from the MgH$_2$(110) surface [98]. This relatively high barrier reflects the slow kinetics of hydrogen release and is in good agreement with experiment [118,119].

Besides surface hydrogen atom desorption, overall dehydrogenation performance of MgH$_2$ also depends on the subsequent hydrogen diffusion from bulk to surface, that is mediated by H-vacancy interactions [120,121]. Surface hydrogen vacancy formation results in the breaking of atomic bonds that affects the surface stability and the dehydrogenation

process [117]. As it was shown by Kurko et al. [121] the increased number of surface hydrogen vacancies reduces the potential barrier for further H-desorption.

The rather high activation barriers for the diffusion of single vacancies from the surface into sublayers (0.45–0.70 eV) suggest that this would easily occur at elevated temperatures [120]; however, these values are essentially lower the activation barrier for hydrogen desorption. It points out that the surface desorption is a restricting factor for hydrogen desorption kinetics in MgH$_2$.

As it was discussed earlier doping with TM atoms improves hydrogen desorption kinetics of MgH$_2$. Impact of different catalysts on hydrogen desorption barrier in MgH$_2$ have been theoretically studied is several works [11,122–126]. Reich et al. [122] studied the role of Ti dopant. The activation barrier was found generally lower as compared to pure MgH$_2$ (up to 14%) but very sensible to the step and doping configuration. As it was shown by Giusepponi and Celino [123] due to Fe atom has a higher coordination than Mg iron doping leads to a local destabilization of the MgH$_2$ lattice that increases probability of hydrogen diffusion towards surface. According to Ri et al. [124] co-substitution by Ti and Fe (adjacent each other at a short distance were found the most energetically favorable) on the MgH$_2$(110) surface results in a large distortion of the lattice and decreases ionicity of H atoms. Compared to single substituted cases such a co-substituted surface is more favorable for hydrogen vacancy formation and leads to further improvement of hydrogen desorption.

An impact of boron doping on hydrogen desorption energies from the MgH$_2$(110) surface was theoretically studied by Kurko et al. [126]. It was found that boron forms strong covalent bonds with hydrogen that perturbs its first and second coordinations and results in both decreasing hydrogen desorption energies and improvement of hydrogen diffusivity. However, boron substitution does not affect much the energy barriers for hydrogen desorption. It is interesting to note that the effect of boron on hydrogen absorption by magnesium is more pronounced: doping with boron leads to a significant decrease in the activation energy of absorption of hydrogen acting as an active center [126].

## 5. Molecular Dynamics Simulation of Mg-H Systems

The main restriction of DFT is the inability to calculate a large number of atoms. Usually, calculations are limited to a few dozen, at best a hundred atoms in a unit cell. For modeling diffusion processes molecular dynamics (MD), classical or using machine learning, is widely applied to gases and liquids, and can be used as an alternative to DFT to study hydrogen diffusivity in metals. MD allows us to simulate large systems (up to tens of thousands of atoms) or long timescale, but requires correct potentials of pair interactions, which can be obtained from ab initio calculations. Considering the interest in magnesium as a material for hydrogen storage, and for other important technological applications, many interatomic potentials have been recently developed for Mg-H systems, including a reactive force field [127], an embedded atom method [128], an angular-dependent potential [129], a bond order potential [130] and most recently a machine-learning interatomic potential based on the Behler–Parrinello approach [131]. Discussion of MD simulation of Mg-H systems is beyond the scope of this review, we only note that MD allows us to describe hydrogen diffusion in a wide hydrogen concentration and temperature ranges [129,131] and also provides an access to the study of such processes as phase separation and formation of magnesium hydride clusters [131]. However, investigations of substitution effects within MD remain problematic, since interatomic potentials for TMs are either absent or need to be corrected to provide satisfactory results [132,133]. Moreover, classical MD fails to describe chemical reactions (dissociation of hydrogen molecules). From this perspective ab initio molecular dynamics, in which dynamic trajectories at a finite temperature are generated using forces calculated from first principles, are more promising. A comprehensive description of the ab initio MD methodology can be found anywhere [134]. In context of the study of hydrogen diffusion in magnesium this method provides very satisfactory results. For example, the hydrogen diffusion coefficient in hcp Mg at 673 K calculated

within ab initio MD by Schimmel et al. [104] ($6.6 \times 10^{-8}$ m$^2$/s) is in a good agreement with both experimental studies [113] and DFT calculations [48] (see Section 4.5).

### 6. Conclusions

This review considers the results of theoretical studies of hydrogen's interaction with Mg obtained within different DFT approaches. DFT proved itself as a powerful method that provides an insight into Mg–H bonding, which, in turn, governs thermodynamics and hydrogen kinetics. DFT allows us to follow pass by pass such processes to which experimental tools are blind or limited: H$_2$ molecules' dissociation of Mg surfaces with further atomic hydrogen absorption, diffusion into and in bulk Mg, hydrogen induced phase transformations in magnesium, hydrogen desorption from MgH$_2$ surfaces and the effect of dopants and vacancies, etc. Due to its accessibility and flexibility, as well as the validity of the results obtained, DFT takes a worthy place among the tools aimed at the design of new materials. In relation to Mg, it allowed us not only to explain the mechanisms responsible for the improvement of MgH$_2$ stability and hydrogen sorption kinetics in the presence of TM additives or other non-metallic dopants, but also to predict new metastable MgH$_x$ phases with better properties, which was further proved experimentally.

**Funding:** The work was supported by Russian Science Foundation (project No. 19-13-00184).

**Conflicts of Interest:** The author declares no conflict of interest.

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
