# Peer review of "Hydrogen Diffusion on, into and in Magnesium Probed by DFT: A Review"

_hydrogen, doi:10.3390/hydrogen3030017_

Round 1
Reviewer 1 Report
Dear Editors,
I have read the paper „Hydrogen diffusion on, into and in magnesium probed by DFT: A review“ and have come up with a few comments/suggestions as stated below. The overall impression is that it is a nice review paper that can be published in “hydrogen” journal after a minor revision.
1. [1]The paper would benefit from proofreading
Line 61 “…that makes it a powerful tool to 60 study hydrogen self-diffusion in metals [27–30], but at a given hydrogen concentration.” “…, but..” this part seems to be disconnected.
Line 65: “…these knowledges..” this knowledge?
Line 85: “…smaller the hydrogen one…” smaller than…?
Etc. …
2. [2] Sec 3.1. “Strictly speaking DFT is not a first-principle (ab initio) method.” I would avoid this discussion in the text since it is not relevant to the topic of the review and contradicts the commonly accepted terminology in the DFT community, which defines DFT calculations as first principles or ab initio. The argumentation that “It depends on how the exchange-correlation functional was constructed, namely with or without any empirical parameters.” is not correct in a general case, as, for instance, the EC functional is known for the free electron gas and a number of systems can be described reasonably well with it without any “empirical parameters”. One can have a discussion of this point in the paper, but I do not see any profit from it in the present manuscript.
3. [3] In Sec 3 and Fig 5 is discussed that ZPE correction is important for H atoms. However, Eq 7 does not have any mentioning of it. Is it included in all energies in the equation there, or is it ZPE correction free? The same concerns results shown in Figs 8-10 and discussed in the text (do they contain ZPE correction or not?) and discussion in line 406 “…ZPE does not contribute much…”, where it would be nice to be quantitative.
Author Response
Dear Reviewer, thank you for your positive feedback and important comments. I hope that by answering your questions, I have improved the article and made it more convenient for the reader. Please find below my answers. All the changes made in the manuscript are marked in red.
- [1]The paper would benefit from proofreading
Line 61 “…that makes it a powerful tool to 60 study hydrogen self-diffusion in metals [27–30], but at a given hydrogen concentration.” “…, but..” this part seems to be disconnected.
It was an attempt to stress that contrary to other methods, e.g. to the permeation technics that studies hydrogen diffusion into a bulk and hence hydrogen concentration changes during experiment, in NMR and QUENS the subject of the study is hydrogen jumps from one interstitial site to another, that results to a translational motion, i.e. diffusion, but the metal-hydrogen system itself is at an equilibrium state, as the hydrogen concentration does not change. To underline the difference between these two diffusion processes, the latter is often called self-diffusion, similar to process of self-diffusion of a liquid in a porous media. I revised the sentence to make it clearer.
Line 65: “…these knowledges..” this knowledge?
It was corrected.
Line 85: “…smaller the hydrogen one…” smaller than…?
Etc. …
It was corrected.
- [2] Sec 3.1. “Strictly speaking DFT is not a first-principle (ab initio) method.” I would avoid this discussion in the text since it is not relevant to the topic of the review and contradicts the commonly accepted terminology in the DFT community, which defines DFT calculations as first principles or ab initio. The argumentation that “It depends on how the exchange-correlation functional was constructed, namely with or without any empirical parameters.” is not correct in a general case, as, for instance, the EC functional is known for the free electron gas and a number of systems can be described reasonably well with it without any “empirical parameters”. One can have a discussion of this point in the paper, but I do not see any profit from it in the present manuscript.
Perhaps you are right, and the present review is not the right place to discuss the family tree of DFT. Moreover, at present it is really customary to refer DFT to ab initio methods, without going into details about how EC functionals were built. I have excluded this paragraph from the text
- [3] In Sec 3 and Fig 5 is discussed that ZPE correction is important for H atoms. However, Eq 7 does not have any mentioning of it. Is it included in all energies in the equation there, or is it ZPE correction free? The same concerns results shown in Figs 8-10 and discussed in the text (do they contain ZPE correction or not?) and discussion in line 406 “…ZPE does not contribute much…”, where it would be nice to be quantitative.
The text has been revised in those places where the indication of the theoretical level applied (accounting for ZPE or not) was omitted. The data reproduced in Figure 10 were calculated with ZPE. In the text it was indicated that activation barriers were calculated with ZPE correction. Further when discussing hydrogen diffusion calculations, I suppose that it is quite clear that these ZPE corrected activation energies were used (moreover, Eq. (4), to which reference is given, comprises ZPE). To mention ZPE once again will look somewhat importunate.
Reviewer 2 Report
The article presents a review of the DFT approaches aimed to the modeling of hydrogen diffusion in magnesium. Strong advantage of the work is a clear description of the calculation methods and their limitations. However, several points need to be addressed:
1. The author didn’t include in the article the calculation of H diffusion coefficient in the hcp Mg for bulk case. This is a little bit strange for a review. In addition, there are several experimental works aimed to the measurement of such coefficients at various temperatures. I recommend to add the figure containing the temperature dependency of H diffusion coefficient in hcp Mg (atomistic simulation and experimental data).
2. It will be great if the author will add a discussion about other DFT-based methods of atomistic modeling of same processes. For instance, the efficient way to describe the behavior of H in metals is the classical atomistic simulation with classical (or machine leaning) interatomic potentials developed with DFT fitting data. Such potentials are available for Mg-H system.
3. The author uses incorrect term “hydrogen self-diffusion”. The term “self-diffusion” means the diffusion of the matrix atoms and cannot be used at description of an impurity diffusion.
4. What is the temperature range where the condition “hv > kT” takes place? I guess it is wrong even at room temperature. If it is true, the author should give more detailed equation for the diffusion coefficient instead eq. (3)
5. English needs to be improved
Consequently, this work may be published after considering the above comments.
Author Response
Dear Reviewer, thank you for your positive feedback and important comments. I hope that by answering your questions, I have improved the article and made it more convenient for the reader. Please find below my answers. All the changes made in the manuscript are marked in red.
- The author didn’t include in the article the calculation of H diffusion coefficient in the hcp Mg for bulk case. This is a little bit strange for a review. In addition, there are several experimental works aimed to the measurement of such coefficients at various temperatures. I recommend to add the figure containing the temperature dependency of H diffusion coefficient in hcp Mg (atomistic simulation and experimental data).
If fact, the calculations of hydrogen diffusion in the hcp bulk were discussed in Section 4.5 with reference to experimental values. But following your recommendations I provided additional references to a review of experimental studies of hydrogen adsorption in Mg and to molecular dynamic simulations.
- It will be great if the author will add a discussion about other DFT-based methods of atomistic modeling of same processes. For instance, the efficient way to describe the behavior of H in metals is the classical atomistic simulation with classical (or machine leaning) interatomic potentials developed with DFT fitting data. Such potentials are available for Mg-H system.
Thank you for this remark, you are right, recently there have been very interesting papers concerning MD-modeling of Mg-H systems. But this is a bit beyond the scope of this review. I added a paragraph (Section 6) just to mention it without going into details.
- The author uses incorrect term “hydrogen self-diffusion”. The term “self-diffusion” means the diffusion of the matrix atoms and cannot be used at description of an impurity diffusion
You are right, and at the same time you are wrong. Indeed, the term “self-diffusion” means the diffusion of the matrix atoms. However, when we deal with hydrogen diffusion in metal-hydrogen systems very often there is a misunderstanding in terminology as depending on experimental technics applied, we study different diffusion processes. E,g, permeation technics studies hydrogen transport diffusion into a bulk, hydrogen concentration changes during experiment, and the diffusion is described by the Fick’s law. in NMR the subject of the study is hydrogen jumps from one interstitial site to another, that results to a translational motion, i.e. diffusion, but the metal-hydrogen system itself is in an equilibrium state, as the hydrogen concentration does not change. The diffusion of H in this case is described using the self-diffusivity, which is related to the motion of a tagged molecule (or atom) by the Einstein relation. And in DFT, if fact, we also study hydrogen self-diffusion. For NMR community this term is widely accepted. In theoretical papers this difference is usually omitted. A comprehensive theoretical study of both transport and self-diffusivity of hydrogen with a lengthy explanation the difference between them can be found in [S. Hao and D. S. Sholl. Self-diffusion and macroscopic diffusion of hydrogen in amorphous metals from first-principles calculations, J. Chem. Phys. 130, 244705 (2009); https://doi.org/10.1063/1.3158619]. I revised the text to underline the difference
- What is the temperature range where the condition “hv > kT” takes place? I guess it is wrong even at room temperature. If it is true, the author should give more detailed equation for the diffusion coefficient instead eq. (3)
In MgH2 phonon frequencies that belong to the vibrations of hydrogen atoms are normally above 600-750 cm-1, See, for example, J. Phys.: Condens. Matter 17 (2005) 7133–7141 or more recent J. Phys.: Conf. Ser. 2145 (2021) 012026. For T < 673 K the condition hv > kT is satisfied. I provided an explanation in the text. Concerning your suggestion to “give more detailed equation for the diffusion coefficient instead eq. (3)” , I’m not sure it is appropriate in this review. I slightly extended the corresponding paragraph and added reference containing more details on this subject.
- English needs to be improved
I tried my best to improve my English.